# Aerosol Delivery of Hesperetin-Loaded Nanoparticles and Immunotherapy Increases Survival in a Murine Lung Cancer Model

**DOI:** 10.3390/nano15080586

**Published:** 2025-04-11

**Authors:** Sayeda Yasmin-Karim, Geraud Richards, Amanda Fam, Alina-Marissa Ogurek, Srinivas Sridhar, G. Mike Makrigiorgos

**Affiliations:** 1Department of Radiation Oncology, Dana-Farber Cancer Institute and Brigham and Women’s Hospital, Harvard Medical School, Boston, MA 02115, USAs.sridhar@northeastern.edu (S.S.); 2Department of Biochemistry, Northeastern University, Boston, MA 02115, USA; fam.a@northeastern.edu; 3Department of Biology, Northeastern University, Boston, MA 02115, USA; aogurek19@verizon.net; 4CaNCURE Program, Northeastern University, Boston, MA 02115, USA; 5Department of Physics, Northeastern University, Boston, MA 02115, USA; 6Department of Bioengineering, Northeastern University, Boston, MA 02115, USA; 7Department of Chemical Engineering, Northeastern University, Boston, MA 02115, USA

**Keywords:** flavonoids, hesperetin, nanoparticles, immunotherapy, aerosol drug delivery

## Abstract

Flavonoids, like Hesperetin, have been shown to be an ACE2 receptor agonists with antioxidant and pro-apoptotic activity and can induce apoptosis in cancer cells. ACE2 receptors are abundant in lung cancer cells. Here, we explored the application of Hesperetin bound to PegPLGA-coated nanoparticles (Hesperetin nanoparticles, HNPs) and anti-CD40 antibody as an aerosol treatment for lung tumor-bearing mice. The Hesperetin nanoparticles (HNPs) were engineered using a nano-formulation microfluidic technique and polymeric nanoparticles. The in vitro studies were performed in human A549 (ATCC) and murine LL/2-Luc2 (ATCC) lung cancer cell lines. A syngeneic orthotopic murine model of lung cancer was generated in wild (+/+) C57/BL6 background mice with luciferase-positive cell line LL/2-Luc2 cells. Lung tumor-bearing mice were treated via aerosol inhalation with HNP, anti-CD40 antibody, or both. Survival was used to analyze the efficacy of the aerosol treatment. The cohorts were also analyzed for body condition score, weight, and liver and kidney function. Analysis of an orthotopic murine lung cancer model demonstrated a differential uptake of the HNPs and anti-CD40 by the cancer cells. A higher survival rate was observed when the combination of aerosol treatment with HNPs was added with the treatment with anti-CD40 (*p* < 0.001), as compared to anti-CD40 alone (*p* < 0.01). Moreover, two tumor-bearing mice survived long-term with the combination treatment, and their tumors were diminished. Subsequently, these two mice were shown to be refractory to the development of subcutaneous tumors, indicating systemic resilience to developing new tumors. Using an inhalation-based administration, we successfully established a treatment model of increased therapeutic efficacy with HNPs and anti-CD40 in an orthotopic murine lung cancer model. Our findings open the possibility of improved lung cancer treatment using nanoparticles like flavonoids and immunoadjuvants.

## 1. Introduction

Lung cancer represents the second most common type of cancer worldwide and the leading cause of cancer death [1], accounting for about one in five of all cancer deaths [2]. Each year, more people die of lung cancer than of colon, breast, and prostate cancers combined [3]. Overall, the chance that a man will develop lung cancer in his lifetime is about 1 in 16; for a woman, the risk is about 1 in 17 [4]. Radiation is an established modality for lung cancer treatment [5]. Radiation induces DNA damage and apoptosis, causing cancer cell fragmentation that exposes cancer-associated antigens to the tumor microenvironment. These can be recognized by antigen-presenting cells (APCs) to induce the antineoplastic effect by activating cytotoxic T cells [6]. In the last few years, there has been a growing interest in cancer immunotherapy due to its promising results in achieving significant and durable treatment responses with manageable toxicity in several cancers, including lung cancer [7]. Our prior studies show that adding immunoadjuvants like anti-CD40 antibodies along with radiation can enhance APC cell activation in their antitumor (M1) format [8]. While beneficial, the use of radiation is also limited by toxicity to normal tissues. For example, when centrally located lung tumors are irradiated, the total radiation dose that can be safely administered is restricted to control radiation toxicity in nearby organs at risk [5].

Flavonoids like Hesperetin, an ACE2 receptor agonist [9,10], can also induce apoptosis in cancer cells [11], whereas ACE2 receptors are overexpressed in lung cancer [12,13]. Hesperetin induces cancer cell apoptosis by targeting multiple biomolecules and the associated signaling pathways, such as ASK1/JNK, p38/MAPK, Notch1, ROS, Bcl-2 family members, and death receptors [14,15,16,17,18]. Accordingly, flavonoid treatment can be used with immunotherapy like anti-CD40 antibody, either as an alternative to radiation or in combination with a moderate amount of radiation. Hesperetin and other flavonoids like Hesperidin, Tramadol, etc., have demonstrated several pharmacological activities, including the anti-COVID-19 effect [10,19,20]. Drug repositioning may facilitate the market availability of these agents if they exhibit anti-cancer activity and get earlier approval as antiviral agents [20,21]. Moreover, nanomaterials or nanoparticles are very popular for their drug delivery and theragnostic effects [22,23,24]. It has also been shown that drug delivery in nanoparticle format enables higher bioavailability and sustainable dispersion with reduced toxicity [22]. Furthermore, the administration of these NPs as a local agent further influences the drug release kinetics and bypasses the fast-pass metabolism and systemic loss [7].

This project aims to evaluate an in vivo inhalation-based drug delivery approach combining nanotechnology with immunotherapy for lung cancer treatment and reduced organ toxicity. We engineered nanoparticles loaded with Hesperetin (HNPs) and employed a setup that delivers the HNPs along with free immunoadjuvant anti-CD40 antibody as an aerosol preparation for inhalation by mice. Key advantages of using this technique include (a) increased bioavailability of Hesperetin, delivered in the form of biodegradable HNPs; while administering free flavonoids can lead to suboptimal bioavailability, nanoparticle-based delivery may partly overcome this limitation [25] and enhance the binding of Hesperetin to lung cancer cell receptors; (b) providing a noninvasive alternative to intravenous (IV) or intramuscular (IM) injections; and (c) in situ delivery of HNPs and anti-CD40 into lung tumors. Due to the large surface area and high vascularization, the lungs are an attractive delivery route for treating respiratory and systemic diseases [26,27], bypassing degradation in the gastrointestinal tract and first-pass metabolism in the liver [27]. Standard intravenous delivery results in less than 1–5% of the drug reaching the tumor [28], while the proposed local delivery is anticipated to boost drug delivery into tumors and improve tumor/non-tumor ratios.

## 2. Materials and Methods

### 2.1. Cell Line and Molecules

Luciferase gene labeled LL/2/Luc-2, from the mouse Lewis lung carcinoma cell line, was purchased from ATCC and was cultured in Dulbecco’s Modified Eagle’s Medium (DMEM) (GIBCO) with 10% FBS (Sigma Aldrich, Burlington, MA, USA) and 1% penicillin/streptomycin (Invitrogen, Carlsbad, CA, USA). Red cherry fluorescent-tagged anti-CD40 was purchased from Abcam, Waltham, MA, USA.

### 2.2. Microfluidics-Based Method to Synthesize PEGylated Nanoparticles

Hesperetin (C_16_H_14_O_6_) and polymers were purchased in powder format from (Sigma Aldrich, Burlington, MA, USA) and kept at 4–80 °C. We used a microfluidic-based synthesis protocol to generate Hesperetin-loaded nanoparticles (HNPs). A syringe pump (Harvard Apparatus Inc., Holliston, MA, USA), microfluidics chips, and tubing (Sigma) were used to synthesize nanoparticles of specific sizes, encapsulating the hydrophobic Hesperetin within biodegradable PEGylated polymers following the company’s instructions. To prepare 1% Hesperetin solution, 1% (*w*/*v*) polymer and 10% (*w*/*v*) Hesperetin were dissolved into dimethyl sulfoxide (DMSO) and acetonitrile (3:2 ratio) as hydrophobic solvents to dissolve the hydrophobic flavonoid (Hesperetin) and polymers. To prepare 1% polymer solution, 1% (*w*/*v*) polymer was used to dissolve into the exact percentages of DMSO and acetonitrile solution. Two polymers were used, PEGPLGA-50L and PEGPLA-L (Sigma), with one block being hydrophilic (polyethylene glycol, PEG) and the other being hydrophobic (poly(lactide-co-glycolide), PLGA, and polylactic acid, PLA). A 1% (*w*/*v*) stabilizer solution, Stabilizer-P (Sigma), dissolves in deionized water. A Harvard apparatus (syringe pump) with two parallel syringe pumps was loaded with a drug or polymer solution for pump one and a stabilizer solution for pump two. The flow rates were adjusted to synthesize nanoparticles of different sizes. Dialysis was performed with 2 L of distilled water to discard the free drug and polymer. The protocol was optimized to generate nanoparticles with varying polymers, PEGPLGA-50L and PEGPLA-L. We synthesized different sizes of NPs using different flow rates (100–470 μL/min) in the presence or absence of Hesperetin in various concentrations (400–800 ng/mL). The prepared nanoparticle morphology was analyzed using a transmission electron microscope under various magnifications at the Egan Research Center at Northeastern University (Boston, MA, USA) using a Joel Jam 1010 Transmission Electron Microscope (TEM) (JEOL, Akishima, Japan). Samples were collected within 24 h after the nanoparticle preparation. The entrapment efficiency (EE, the ratio between encapsulated over initial drug amount) and loading capacity (total entrapped drug divided by total weight of the polymer) of the prepared HNPs were analyzed. The Hesperetin-loaded nanoparticle solutions were centrifuged using a centrifugal filter (8 K–10 K cutoff, MilliporeSigma, Burlington, MA, USA). A known amount of this sample and free drug was completely dissolved in dimethyl sulfoxide (DMSO, Sigma-Aldrich), and then the drug absorbance was measured at 320 nm using a UV–vis spectrophotometer (Agilent BioTek Gen5™ Microplate Reader, Agilent Technologies, Santa Clara, CA, USA). The HNP concentrations were calculated from the absorbance using a previously established calibration curve. Loading capacity (LC) was calculated by the amount of total entrapped drug divided by the total polymer weight, as described [29,30]. All the prepared nanoparticles were kept at 4–8 °C and used within 30 days of preparation.

### 2.3. Clonogenic Assay

For clonogenic assays, A549 cells were first cultured and we then followed protocols described earlier [31,32,33]. A 100 cells/well concentration was seeded in 12-well plates with 1 mL of the mixture per well. After 48 h seeding, the cells were treated with Hesperetin or the vehicle. Plates were then allowed to incubate for 2–3 weeks at 37 °C with 95% air and 5% carbon dioxide, where the media were changed for all the plates every 48–72 h. Once an endpoint was determined via adequate growing colonies of >50 cells/colony in the control plates, the media were removed from all the plates, and the cells were fixed with 75% ethanol in PBS and stained with 0.5% crystal violet. The plates were then washed gently using water to remove the crystal violet stain and then imaged via a flatbed scanner and digitized via an imager for colony counting and analysis. The data are the mean of three independent experiments.

### 2.4. Computed Tomography (CT) Imaging

CT imaging was performed using a small animal radiation research platform (SARRP, Xtrahl, Inc., Suwanee, GA, USA), as described [34]. The mice were anesthetized with isoflurane vapor, and CT images were taken at 65 kVP X-ray energy.

### 2.5. Pathology and Histology

Ex vivo tissues were collected and fixed with 10% formalin for 24 h. The tissues were processed for paraffin embedding by iHisto Co. (Salem, MA, USA) for gross pathology to identify lung tumor prognosis. For hematoxylin and eosin (H&E) staining, the paraffin-embedded tissues were sliced into four µm-thick sections with a microtome, air-dried, fixed with acetone, and stained following the standard protocol. The sections were stained with hematoxylin and eosin (H&E) to observe lung metastasis, and whole-slide scanning (40×) was performed on an EasyScan infinity (Motic Digital Pathology, Emeryville, CA, USA). High-magnification images were collected using Case Viewer software (version 2.4.0.119028).

### 2.6. Mouse Lung Tumor Models

C57BL/6NTac background 8–12 week-old wild (W+/+) male and female mice were purchased from Taconic mice. The animals were contained in groups of five in standard cages with free access to food and water and a 12 h light/dark cycle. All the mice were adjusted to the animal facility for at least 1 week before experimentation. All possible parameters that may cause social stress, like group size, among the experimental animals were carefully checked and evaded. The animals were observed three times a week after cell implantation for any physical abnormalities. To better mimic the lung cancer scenario, orthotopic lung tumors were generated by implanting cancer cells in the mouse lungs using an endotracheal intubation kit (Kent Scientific Kit, Kent Scientific Corporation, Torrington, CT, USA). C57BL/6 background mice were implanted with the matched background LL/2-Luc2 (10 × 10^5^ cells/tumor) luciferase-positive cell line (ATCC). Luciferase-expressing mouse cells are ideal for in vivo bioluminescence imaging to monitor NSCLC tumor development in vivo.

For subcutaneous (SQ) tumor models, LL/2-Luc2 (5 × 10^4^ cells/tumor) suspended in PBS were injected into one of the mice’s flanks. In all cases, an insulin syringe with a 22-gauge needle was used for subcutaneous tumor cell injection. All the animal experiments were conducted in compliance with the guidelines and regulations set by the Dana Farber Cancer Institute (DFCI, Boston, MA, USA) Animal Care and Use Committee (IACUC).

### 2.7. Aerosol Drug Delivery

We developed a mouse aerosol drug delivery system using a hospital-grade Schuco air compressor, a mouse drug delivery mixing chamber, and a mouse pie cage for a nebulizing drug delivery system (Braintree Scientific, Inc., Braintree, MA, USA). The mouse pie case is designed for aerosol drug delivery to up to 10 mice at a time. The aerosol enters the pie via tubing attached to the central core. The core has a sealable opening apparatus on the top for the mouse placement and a solid bottom. The aerosol exits each pie section through one 3 mm hole in the center of the outer edge, through which the aerosol drugs can be dispersed to ten chambers at the same rate. We followed the recommendations supplied by the manufacturer (Braintree Scientific Inc., Braintree, MA, USA), and we found that the recommendations resulted in successful experimental outcomes. The recommended settings for the nebulizer compressor are a compressor pressure of 35–45 psig (pound per square inch gauge), with an operating flow rate of 7–11 Ipm (inches per minute), which is appropriate for releasing therapeutic particles of sizes 0.5 μm–5 μm using 120 V AC, 60 Hz. Using this aerosol drug delivery system, we successfully treated lung tumor-bearing mice with one ml of prepared HNP (PEGPLGA-50L, L size) solution (500 ng/mL concentration of Hesperetin) and/or 1 mL of free anti-CD40 (40 µg/mL concentration) diluted to 5 mL with sterile ddH_2_O, and we used the total volume for each time aerosol treatment. We carried out the treatment for 30 min each time for a total of 3 times with an interval of 3 days. An estimated ~2500 ng/kg of Hesperetin was delivered per mouse in HNP aerosol format. We used the same type and sized drug-free polymer NPs for the control mice with a similar dilution. We used freshly prepared anti-CD40 solution in each treatment and never stored the prepared solution. For all the cohorts, body weight and body condition scores were measured. Body conditions were scored following the DFCI IACUC guidelines (one being the lowest and four being the highest score based on body fat condition).

### 2.8. Tumor Rechallenge Study Design

Four naive 16-week-old C57BL/6NTac background wild (W+/+ mice (Taconic Bioscience, Germantown, NY, USA) were implanted with SQ tumors, along with two surviving mice that had achieved tumor remission (HNP- + anti-CD40-treated). For subcutaneous (SQ) implants, LL/2-Luc2 (5 × 10^4^ cells/tumor) suspended in PBS was injected into one of the mice’s flanks. In all cases, an insulin syringe with a 22-gauge needle was used for the subcutaneous tumor cell injection. The tumors were monitored for four weeks post-implant. On days 7–9, palpable tumors of a size comparable to a pea were developed in the naive mice. All the mice were observed every week until week 4, when the tumor in the naive mice reached its endpoint; however, no tumor was palpable in the rechallenged mice. We performed a CT image of all six mice (4 naive and 2 rechallenged) before euthanizing the naive mice. We also had to euthanize our tumor-regressed, rechallenged mice to follow our animal protocol. All the animal experiments were conducted in compliance with the guidelines and regulations set by the Dana-Farber Cancer Institute (DFCI) Animal Care and Use Committee (IACUC).

### 2.9. Statistical Analysis

The survival data were plotted, and statistical analyses were performed using GraphPad prism v7.0. A log-rank test was employed to determine the *p*-value for the Kaplan–Meier curves. The statistical analyses for tumor volume were achieved using two-way ANOVA: two factors with the replication tool and a standard Student’s two-tailed *t*-test. *p*-values < 0.05 were considered significant. * *p*  <  0.05, ** *p*  <  0.01, *** *p*  <  0.001, and **** *p*  <  0.0001 were considered as statistically significant at the 95% confidence interval.

## 3. Results

We first analyzed the effect of Hesperetin (Figure 1a) on the proliferation of the human A549 lung cancer cell line, using a clonogenic survival assay, which showed a significant decrease in cell survival with 5 (*p* < 0.01), 15 (*p* < 0.001), and 25 (*p* < 0.0001) µg/mL of the Hesperetin treatment doses for 24 h, compared to the controls. Here, the cell survival percentage decreased in a dose-dependent manner, with an IC_50_ value of approximately 10 μg/mL concentration of Hesperetin (Figure 1b). Next, given the evidence of the anti-cancer cell activity of Hesperetin towards the human lung cancer cell line, using a microfluidic technology (Appendix A), we generated Hesperetin-loaded nanoparticles (HNPs) of two different sizes, smaller (S) and larger (L) (Figure 2a,b and Appendix A). The sizes of the HNPs were measured from TEM images taken by an electron microscope. The particles look spherical and well contoured. Adding the Hesperetin to the polymer-based nanoparticles was anticipated to improve the bioavailability of the flavonoid in tissues via slow and sustained in situ delivery of the drug payload [35]. To analyze the HNPs’ effect on lung cancer treatment, we used an in vivo murine subcutaneous (SQ) lung cancer model by generating tumors in immunocompetent C57BL/6 mice with a matched background mouse lung cancer cell line (LL/2-Luc2). Here, we used HNPs of two different sizes using two different polymers, PEGPLGA-50L (Sigma) and PEGPLA-L, in the larger-sized [L] (average diameter 60 nm, range = 37–105 nm) and smaller-sized [S] (average diameter 40 nm, range = 31–60 nm) HNPs, respectively. Palpable-sized SQ tumor-bearing mice were randomized to different cohorts, and solutions of the HNPs were injected directly into the tumor (intratumorally, IT) for treatment. Two different concentrations of HNPs, 400 ng/mL (200 ng/kg) and 800 ng/mL (400 ng/kg) of the HNP solutions, for each size (S and L) were used for the different cohorts. The total treatment was given three times, three days apart (Figure 2c). Two different sizes of HNPs (S with PEGPLA-L and L with PEGPLGA-50L polymers) were used for each concentration of HNPs. The SQ tumor volume data show that HNPs can suppress the growth of lung tumors in mice in a dose-dependent manner; however, at low doses, the L-sized HNPs with PEGPLGA-50L polymers were found to be more effective (Figure 2d). Thus, for subsequent studies, we used these L-sized HNPs. Our analysis of these larger HNPs reveals an EE of 0.2% and an LC of 0.4%, under 1% polymer concentration and 2.5% drug/polymer (*w*/*w*) ratio.

To further evaluate the in vivo action of HNPs in murine lung cancer, we generated a lung orthotopic mouse cancer model and followed the growth with BLI imaging (Figure 3a). We used intubation to implant the luciferase reporter-labeled cells LL/2-Luc2LLC-1 on the same C57BL/6 mouse background lung cancer cell line to generate orthotopic lung cancer, with one tumor per mouse (Appendix A). We then employed an aerosol drug delivery system using a nebulizer (Figure 3b and Appendix A) to establish an inhalation drug delivery system for lung cancer treatment. Using this aerosol inhalation system, we initially employed green fluorescence polymer nanoparticles (NPs) to confirm the uptake of NPs by lung tissue when using the aerosol drug delivery system. In addition, we used red cherry fluorescence-tagged anti-CD40 for the inhalation treatment along with the green fluorescence polymer NPs. To observe the uptake in tumor-inoculated lungs, we used lung orthotopic tumor-inoculated mice and healthy mice (Figure 3c). The tumor formation was initially observed via BLI imaging and later confirmed by ex vivo H&E images of the resected lungs (Figure 3d). Two weeks after the tumor inoculation, both groups of mice were given aerosol delivery of green fluorescence-tagged polymer NPs and red fluorescence-tagged anti-CD40 for 30 min simultaneously. Ex vivo resection of the lung’s frozen section (Figure 3e) within an hour following the treatment demonstrated preferential uptake of both HNPs (*p* < 0.01) and anti-CD40 (*p* < 0.001) (Figure 3f,g) by a graph and corresponding images of lung tumor tissues (lower panels) as compared to healthy lung tissues (upper panels).

Following confirmation of the nanoparticle and anti-CD40 uptake in the lung tumor cells, we conducted experiments to observe the therapeutic effect of HNPs along with free anti-CD40 following aerosol drug delivery (Figure 4a). The data demonstrate an increased survival rate and duration in the case of tumor-bearing mice treated with free anti-CD40 inhalation compared to the placebo group (Figure 4b). Further survival enhancement was observed when HNPs were added along with anti-CD40 during aerosol therapy, as shown by the Kaplan–Meier survival graph (Figure 4b). Representative BLI images in Figure 4c show day 0 of treatment starting in the upper panel and the specific evaluation days in the lower panel. Ex vivo H&E staining of the lung was performed in a fraction of the mice, showing tumor regression in representative images (Figure 4d).

Two of the nine treated mice with combination therapy demonstrate complete tumor regression with a more than 300-day post-treatment survival duration. A subsequent rechallenge of these two long-surviving mice was performed with subcutaneous tumor cell implantation using the same LL/2-Luc2 cell line (Figure 4e), and no tumor growth was observed in the surviving combination-treated group. In contrast, all four naive mice with subcutaneously implanted LL/2-Luc2 cells developed SQ tumors as shown in manually (with a digital caliper) measured tumor diameter (Figure 4f) and by CT images (Figure 4g). The data are consistent with a “vaccine effect” development in the two long-surviving mice from the combination treatment group.

The toxicity study demonstrates no change in body score and body weight in the treatment cohorts (Figure 5a,b) of the tumor-bearing mice, and no significant deviation was found in the liver and kidney treatment for the aerosol HNP treatment when checking after three hours (Figure 5c) and three days post-treatment (Figure 5d) in the non-tumor-bearing mice.

## 4. Discussion

Our ability to manipulate and manufacture materials on the nanoscale level has allowed us to develop nanotechnology–based approaches combined with radiation therapy, chemotherapy, or immunotherapy. In earlier work, we proposed the replacement of fiducials used in radiation therapy for guiding radiation beams to the tumors with gold nanoparticles loaded with radio sensitizers, encapsulated in slow-release polymers, to enable both beam localization and biological sensitization [34,35]. This approach can be used both for uniform, externally applied radiation therapy as well as with non-uniform radiation therapy delivered via brachytherapy or internal emitter radiation therapy [35,36,37,38,39,40,41]. In the present work, we employed nanotechnology-based drug inhalation strategies for improving therapeutic efficacy and mitigating adverse side effects in lung cancer-harboring mice. Polymeric nanoparticles remain stable during nebulization and induce minimal toxicity, while having tunable degradation rates in vivo [22]. This method of drug delivery is anticipated to be advantageous, and recent systematic reviews demonstrate that NP inhalation therapies achieve higher drug deposition in the lungs than conventional methods [42]. Here, our experimental protocol proved effective in introducing Hesperetin onto nanoparticles, enabling the inhalation of the nanoparticles, and eventually releasing Hesperetin in lung cancer tissues where, along with free immunoadjuvant anti-CD40, they reduced or eliminated the growth of implanted lung tumors.

Based on our efficacy study analysis, we observed the functionality of these NPs in a lung cancer model as an aerosol drug delivery method. In general, the development of NPs displays different mechanical properties compared to bulk materials. Thus, changes in some of the physicochemical properties, like mechanical, electronic, optical, or catalytic properties, may significantly enhance the reactivity and selectivity of HNPs compared to their bulk analogs [43]. This enhancement may further influence the cellular uptake and reactivity toward lung cancer and other cancer cells and tissues. A detailed analysis will reveal these characteristic changes in the HNPs compared to their bulk material as an aerosol dispersion. Regarding the HNP sizes used for the SQ lung cancer model, we observed a non-significant variation because of size at higher doses. In the case of low doses, the size difference is evident, while a larger size demonstrates higher efficacy and has higher drug capacity. Accordingly, we decided to use the larger HNPs for aerosol delivery in this study, in vivo. While the EE of Hesperetin in nanoparticles can probably be improved further, the focus of this study was to provide proof of principle for treatment effectiveness; hence, detailed EE optimization studies were not conducted.

Lung cancer evades the immune response through multiple mechanisms. For instance, lung cancer cells may undergo immunoediting, in which precancerous cells slowly undergo selective adaptation to become “invisible” to immune surveillance [23]. Previous studies [25,44], including work from our group [6,7,8], demonstrated that immunoadjuvants like anti-CD40 induce antitumor effects in different cancer models, including the lung, while apoptosis induced by radiation initiates further enhancement of immune cell activation [6,7]. Here, we used Hesperetin as a pro-apoptotic agent [12], instead of radiation. We show that adding Hesperetin-loaded nanoparticles to the immunoadjuvant anti-CD40 and delivery in an aerosol format is effective in treating orthotopic lung cancer in mice, even without the inclusion of radiation. In addition, there is evidence that a fraction of the mice (2/9) obtained long-term immunity that prevented subcutaneous tumor formation upon rechallenging with the same tumor cell line. The data are consistent with an abscopal, cancer vaccine effect [26], upon which endogenous T cells are primed against the cell line tumor antigens following the applied aerosol treatment. While the abscopal effect hypothesis must be validated with additional studies, almost all the treated mice exhibited an increase in survival following a relatively simple inhalation treatment that was administered noninvasively, and without advanced equipment. This treatment approach may potentially allow lung cancer treatment that is readily available in under-resourced environments that have reduced access to radiation and without the expense and hazards associated with radiation treatment. Alternatively, the treatment could be offered in conjunction with fewer radiation courses and a reduced total radiation dosage. We anticipate that nanoparticle-based aerosol drug delivery locally for lung cancer, potentially in conjunction with locally administered radiation, may enhance the overall efficacy of treatment while drastically reducing systemic treatment complications.

## 5. Conclusions

We successfully developed an aerosol drug delivery model to administer both anti-CD40 and HNPs in a murine lung cancer model. Our analysis of an orthotopic murine lung cancer model demonstrates a preferential uptake of the HNPs and anti-CD40 by the cancer cells, sparing the normal lung tissue. Moreover, the highest survival rate was observed with the combination aerosol treatment with HNPs + anti-CD40, compared to anti-CD40 alone or HNPs alone. Our findings open the possibility of improved lung cancer treatment using flavonoids and immunoadjuvants, using a minimally invasive approach for therapy delivery. This treatment model could potentially allow lung cancer treatment that is readily available for the mass population without the side effects of radiation treatment. Further work is needed for detailed characterization and to identify the mechanism(s) of action of these HNPs in lung cancer treatment as an aerosol-administered drug.

## Figures and Tables

**Figure 1 nanomaterials-15-00586-f001:**
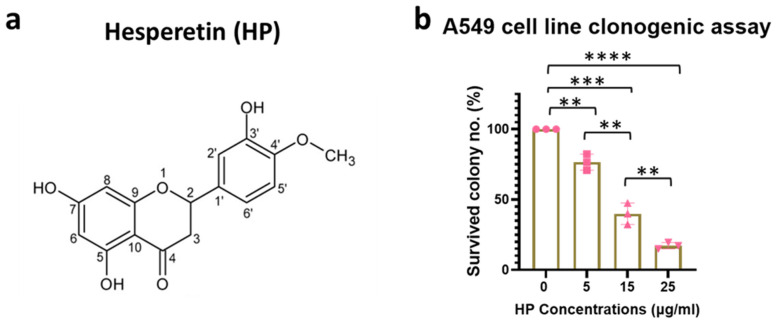
Antitumorigenic effect of Hesperetin in human lung cancer cell line. (**a**) Biochemical structure of Hesperetin. (**b**) Bar graph showing percent colony count for the clonogenic survival assay (n = 3) in human lung cancer cell line, A549, treated with Hesperetin (Sigma) doses from 5 to 25 µg/mL for 24 h. Data represent the mean +/− standard deviation (SD) of three experiments. ** *p* < 0.01, *** *p* < 0.001, **** *p* < 0.0001 at 95% confidence interval. The symbol “n” is for the sample number.

**Figure 2 nanomaterials-15-00586-f002:**
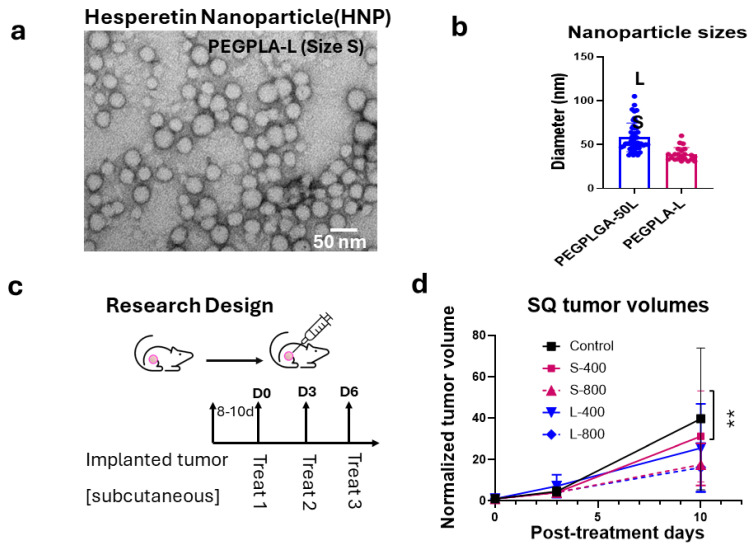
Preparation of Hesperetin-loaded nanoparticles (HNPs) and efficacy in a subcutaneous (SQ) mouse tumor model. (**a**) A transmission electron microscopy (TEM) image of HNPs (representative of size S) and (**b**) Bar graph of the different-sized (S and L representing 31–60 nm and 37–105 nm in diameter, respectively) HNPs used in the efficacy study in the SQ lung tumor model. (**c**) Study design and workflow for the in vivo study. (**d**) Tumor growth curve, depicting fold increase in tumor volume relative to tumor volume at day of treatment initiation. This is shown for different sizes and concentrations of HNP treatment in an SQ mouse model for mouse lung cancer showing up to 10 days while control mice were still alive. A single SQ cancer was implanted in one of the flanks of C57BL/6 background immunocompetent mice with matched background mouse LL/2-Luc2 cell line, and all cohorts (n = 5) were treated when the tumor reached palpable size. Each HNP size was injected intratumorally for treatment, separately for lower and higher doses. Control mice were treated with the same vehicle volume (100 µL). Data represent the mean +/− SD. ** *p* < 0.01 at 95% confidence interval.

**Figure 3 nanomaterials-15-00586-f003:**
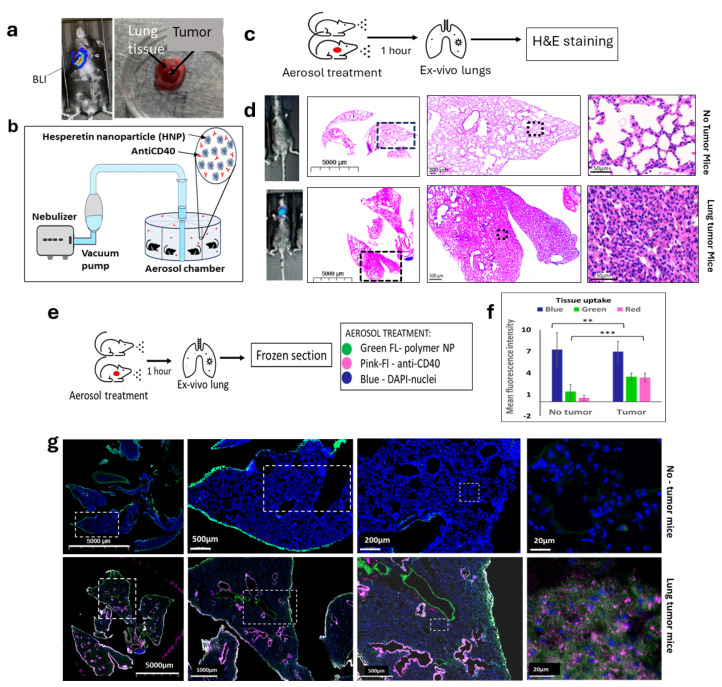
Establishment of aerosol delivery of HNPs and anti-CD40 in lung cancer orthotopic animal model. (**a**) BLI image (**left**) of a mouse showing the development of lung tumor by the implantation of mouse lung cancer cells (LLC1) under the guidance of endotracheal intubation tubing. (**Right**) ex vivo picture of tumor grown in lung tissue. (**b**) Schematic diagram of the aerosol drug delivery using a nebulizer vacuum pump connected to a drug mixture chamber delivering to an aerosol drug delivery chamber. (**c**) Treatment design for the aerosol drug delivery of HNPs and free anti-CD40 in mice followed by hematoxylin and eosin (H&E) staining in ex vivo lung tissue in naive and tumor bearing mice. (**d**) Bioluminescence (BLI) images of the mice and H&E staining images of the ex vivo lung tissue of mice under different magnifications show no tumor (**upper panel**) and lung tumor-bearing mice (**lower panel**). Black dashed frame area of each picture represents the full picture of the next picture. (**e**) The research design for aerosol drug delivery of FITC-labeled polymer NPs (green, wavelength~350 nm) and red fluorescently tagged anti-CD40 (red, web length of ~500 nm). Cells are represented by DAPI (blue) staining of the nuclei. (**f**) The bar graph and (**g**) representative fluorescence images of ex vivo lung tissue showing cell nuclei stained by DAPI (blue), fluorescent polymer NP (green) uptake, and red cherry-tagged anti-CD40 antibody uptake (hot pink). White dashed frame area of each picture represents the full image of the next picture. Diffuse co-localization of fluorescent nanoparticles and anti-CD40 was observed in tumor-bearing mice (**lower panel**) but significantly lower in the tumor-free mice (**upper panel**) in tumor microenvironment. Data represent the mean +/− standard deviation (SD) of three experiments. ** *p* < 0.01, *** *p* < 0.001 at 95% confidence interval.

**Figure 4 nanomaterials-15-00586-f004:**
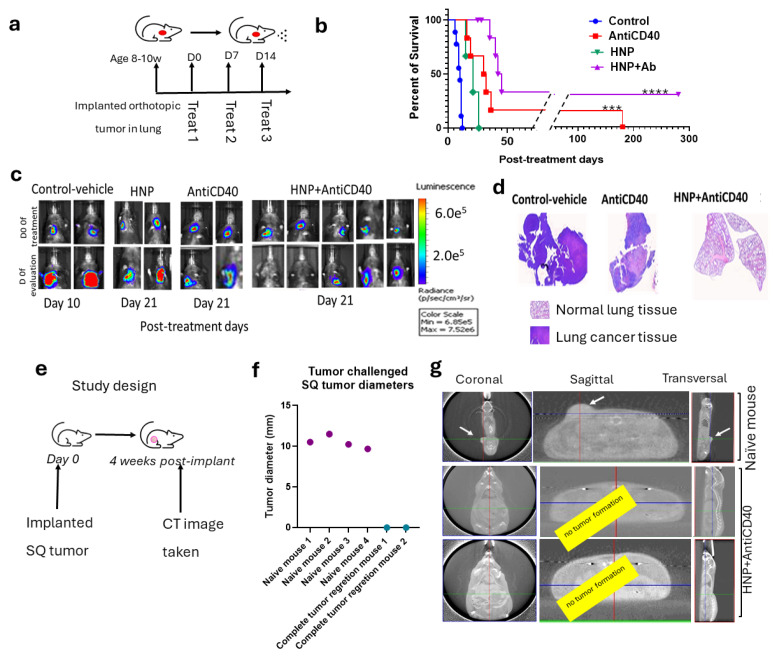
Efficacy of HNPs and anti-CD40 in lung cancer orthotopic animal model with aerosol drug delivery. (**a**) A schematic diagram of the aerosol drug delivery in orthotopic lung cancer bearing mice. (**b**) Kaplan–Meier survival graph showing the survival percent and duration of different cohorts of treated mice with aerosol delivery of anti-CD40 (n = 6), HNPs alone (n = 3), HNPs + anti-CD40 (n = 9), and control (n = 8). (**c**) Representative BLI images of the different cohorts from the treated mice with controls. The upper panel represents day 0 of the treatment, and the lower panel represents the post-treatment time for different cohorts: control (day 10), anti-CD40 (day 21), HNPs (Day 21), and HNPs + anti-CD40 (day 21). (**d**) Representative H&E images from the ex vivo lungs confirm some of the corresponding treatment outcomes. (**e**) Study design. (**f**) The graph shows the outcome of the tumor-rechallenged study with an SQ implant of LL/2-Luc2 tumor cells of the HNP- + anti-CD40-treated tumor-recovered mice (n = 2) with a cohort of naive mice as control (n = 4). (**g**) CT images were taken four weeks post-injection of tumor cells, confirming the outcome with tumor growth in naive and HNP + anti-CD40-treated mice with complete tumor regression. For statistical significance at a 95% confidence interval *** *p* < 0.001, **** *p* < 0.0001.

**Figure 5 nanomaterials-15-00586-f005:**
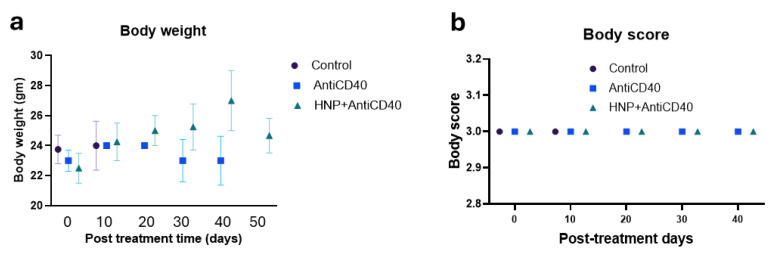
Toxicity study. (**a**) Body weight and (**b**) body scores were measured during the treatment period of the tumor-bearing mice for all cohorts of the aerosol-treated mice along with the controls. Each data represents the average outcome of the cohort for that time. (**c**) Three hours (n = 5) and (**d**) three days (n = 4) post-treated liver and kidney function clinical reports of the combination-treated (HNPs + anti-CD40) mice with aerosol-treated doses in wild naive (non-tumor bearing) mice. Ex vivo serum was collected for clinical laboratory reports for analysis of serum alkaline transaminase (ALT, normal range 15–80.1 U/L), serum aspartate transaminase (AST, normal range 33.95–268.47 U/L), serum bilirubin (normal range 0.17–0.53 mg/dL), and serum creatinine (normal range 0.12–0.43 mg/dL). Each piece of data represents an individual mouse. Red font arrows indicate the mouse’s upper and lower range of normal serum levels.

## Data Availability

Data are contained within the article or Appendix A. Data are also available on request from the corresponding author.

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
