# Peer review of "Aerosol Delivery of Hesperetin-Loaded Nanoparticles and Immunotherapy Increases Survival in a Murine Lung Cancer Model"

_nanomaterials, 2025, doi:10.3390/nano15080586_

Round 1
Reviewer 1 Report
Comments and Suggestions for Authors
Karim et al. propose the potential for improved survival through the aerosol delivery of hesperetin-loaded nanoparticles (HNPs) combined with immunotherapy in a murine lung cancer model. The authors first confirm the tumor-suppressive effects of hesperetin at the cellular level. Subsequently, they encapsulate hesperetin within nanoparticles (HNPs) and administer them directly into subcutaneous (SQ) tumors in murine models, demonstrating tumor inhibition. To assess in vivo uptake via aerosol delivery, they employ an orthotopic mouse cancer model, ultimately evaluating the therapeutic efficacy of HNPs in combination with immunotherapy. Additionally, they examine the vaccine-like effect of this treatment through an SQ tumor rechallenge.
This study is significant in two key aspects. First, it highlights the advantages of aerosol-based drug administration over conventional intravenous delivery, potentially minimizing systemic side effects. Second, it demonstrates the supplementary effects of hesperetin in enhancing immunotherapy outcomes without additional adverse effects.
However, several aspects require further clarification and improvement. First, the nebulizer-based delivery system used in the study may pose limitations in controlling drug concentration. Given the critical role of precise drug dosing in anticancer treatment, additional experiments or explanations are necessary to evaluate how chamber concentration and nebulizer flow rate influence in vivo uptake. Second, while the study systematically varies nanoparticle size and drug concentration in direct intratumoral injections, the aerosol-based treatment lacks such variations. Further experiments or a discussion on this discrepancy would strengthen the study. Third, in the final survival analysis, two murine models exhibited prolonged survival (>300 days) before undergoing an SQ tumor rechallenge, demonstrating a vaccine-like effect. However, the study does not provide an evaluation of the duration of this effect. Additionally, in Figure 4F, the comparison with the four untreated mice lacks details on the timeline of tumor implantation and evaluation. Furthermore, there is a typographical error in Figure 4F ("tumor chalanged" should be corrected to "tumor challenged").
Author Response
Reviewer 1
Comment 1: First, the nebulizer-based delivery system used in the study may pose limitations in controlling drug concentration. Given the critical role of precise drug dosing in anticancer treatment, additional experiments or explanations are necessary to evaluate how chamber concentration and nebulizer flow rate influence in vivo uptake.
Response 1: The reviewer is correct, that one may perform additional studies to optimize the flow rate for optimal uptake. While we plan to perform such studies in the future, in the current work, we followed the recommendations supplied by the manufacturer (Braintree Scientific), and we found that the recommendations resulted in successful experimental outcomes. The recommended settings for the Nebulizer Compressor are a compressor pressure of 35-45 psig (pound per square inch gauge), with an operating flow rate of 7-11 Ipm (inch per minute), which is appropriate for releasing therapeutic particle sizes 0.5 μm-5μm using 120V AC, 60 Hz. We have now added this information to the methods section of the manuscript (Lines: 171-196).
Comment 2: Second, while the study systematically varies nanoparticle size and drug concentration in direct intratumoral injections, the aerosol-based treatment lacks such variations. Further experiments or a discussion on this discrepancy would strengthen the study.
Response2: We appreciate the reviewer’s suggestion. We added a few lines on this in the discussion section (line 405-407) and the following (reference #43) :
Reference: Cojocaru, Elena, Ovidiu Rusalim Petriș, and Cristian Cojocaru. 2024. "Nanoparticle-Based Drug Delivery Systems in Inhaled Therapy: Improving Respiratory Medicine" Pharmaceuticals 17, no. 8: 1059. https://doi.org/10.3390/ph17081059”
Comment 3: Third, in the final survival analysis, two murine models exhibited prolonged survival (>300 days) before undergoing an SQ tumor rechallenge, demonstrating a vaccine-like effect. However, the study does not provide an evaluation of the duration of this effect.
Response 3: While it would be very interesting to perform a systematic evaluation of the duration of the vaccine-like effect, this was beyond the scope of this study. Our institutional animal facility protocol has a time limit of 300 days for observing survival. Observing the vaccine effect beyond 300 days was not possible under these circumstances. We will keep this suggestion in mind for a subsequent study, after amending the protocol.
Comment 4: Additionally, in Figure 4F, the comparison with the four untreated mice lacks details on the timeline of tumor implantation and evaluation.
Response 4: We did include a brief description of subcutaneous tumor implantation for the tumor rechallenge in the study design subsection: 2.6 Mouse and tumor models. We have also added more details in the methods section and a new subheading: 2.8 Tumor Rechallenge Study Design. Also, Figure 4E shows the timeline diagrammatically.
Comment 5: Furthermore, there is a typographical error in Figure 4F ("tumor chalanged" should be corrected to "tumor challenged").
Response 5: This typo has been corrected.
Reviewer 2 Report
Comments and Suggestions for Authors
Yasmin-Karim and co-workers are reporting the preclinical therapeutic outcomes of PEGylated nanoparticles for aerosol delivery of Hesperet in a murine lung cancer model. This research study is interesting and can make a significant contribution to the field of drug delivery and cancer therapy. However, authors should address the following minor and major revisions before the manuscript can be considered for publication in nanomaterials.
- Every word in the title must start with an alphabet.
- The subsections under section 2 should be numbered as 2.1., 2.2., and so on.
- The country of manufacturing and model number of the transmission electron microscope (TEM) instrument used for particle size analysis must be mentioned.
- The TEM analysis for characterization is not enough, the authors should also consider adding other characterization techniques such as FTIR, SEM, etc.
- The particle sizes of nanoparticles from TEM must be mentioned in the results as well, not only in Figure 2. Moreover, the significance of the obtained particle sizes of nanoparticles must discussed in the discussion section
- The ‘’50” of IC50 should be in subscript in line 212.
- The entrapment efficiency and loading capacity results should be included in section 3 and explained accordingly under section 4.
- The quantities of reagents that were used for the preparation of Hesperet-loaded nanoparticles should be mentioned in section 2.
- The authors must also reduce the %similarity of the manuscript, it is too high (about 67%).
Author Response
Reviewer 2
Comment 1: Every word in the title must start with an alphabet.
Response 1: The title has been updated according to the suggestion.
Comment 2: The subsections under section 2 should be numbered as 2.1., 2.2., and so on.
Response 2: The subsections have been numbered accordingly.
Comment 3: The country of manufacturing and model number of the transmission electron microscope (TEM) instrument used for particle size analysis must be mentioned.
Response 3: The information has been added to the method section: 2.2 Microfluidics-based method to synthesize PEGylated nanoparticles (line 126-127)
Comment 4: The TEM analysis for characterization is not enough, the authors should also consider adding other characterization techniques such as FTIR, SEM, etc.
Response 4:The recommendation of the reviewer is appreciated. Unfortunately our Core Facility lacks Fourier transform infrared spectroscopy (FTIR) and Scanning Electron Microscopy (SEM), hence we have not been able to perform these extra analyses. FTIR is used to identify functional groups present in organic and inorganic compounds. Here, the chemical composition of our drug is known. Further, SEM produces detailed, magnified images of an object by scanning its surface to create a high-resolution image. While this would be useful to have, here we already know the chemical ingredients of this nanoparticles.
Comment 5: The particle sizes of the nanoparticles, as determined by TEM, should be mentioned in the results, not just in Figure 2. Moreover, the significance of the obtained particle sizes of nanoparticles must discussed in the discussion section
Response 5: The particle sizes of nanoparticles from TEM are mentioned in the result section (lines 242-243 and 250-251).
The significance of the obtained particle sizes of nanoparticles has also been discussed in the discussion section (lines 420-424).
Comment 6: The ‘’50” of IC50 should be in subscript in line 212.
Response 6: We updated the ‘50’ to be a subscript.
Comment 7: The entrapment efficiency and loading capacity results should be included in section 3 and explained accordingly under section 4.
Response 7: This information has been added in the result section, section 3, (Lines: 259-262). A paragraph is also added in the discussion section, section 4, Lines:424-427.
Comment 8: The quantities of reagents that were used for the preparation of Hesperetin-loaded nanoparticles should be mentioned in section 2.
Response 8: This information has been added in subsection 2.2, Microfluidics-based method to synthesize PEGylated nanoparticles (lines 109-119).
Comment 9: The authors must also reduce the %similarity of the manuscript, it is too high (about 67%).
Response 9: we are not aware of another publication with 67% similarity. However, we did publish a pre-print in Bioarxiv, if this is what the reviewer refers to. In our understanding, our own pre-prints should not be considered in the % similarity for a peer reviewed publication.
Reviewer 3 Report
Comments and Suggestions for Authors
1、Replace "ml" with "mL" throughout the manuscript to adhere to proper scientific notation.
2、Line 246 contains two consecutive periods. Please correct this punctuation error.
3、The term "Hesperetin" is misspelled as "Hesperein" in multiple instances. Ensure consistent and correct spelling throughout the manuscript.
4、The reference formatting is inconsistent. Please revise all references to comply with the journal's guidelines.
5、The manuscript lacks detailed nanoparticle synthesis parameters, including encapsulation efficiency (EE) and loading capacity (LC) values. These should be provided to enhance the credibility and reproducibility of the experiments.
Author Response
Reviewer 3
Comments and Suggestions for Authors
Comment 1: Replace "ml" with "mL" throughout the manuscript to adhere to proper scientific notation.
Response 1: Thanks for the advice. This has been updated.
Comment 2: Line 246 contains two consecutive periods. Please correct this punctuation error.
Response 2: Thanks for the advice. This has been corrected.
Comment 3: The term "Hesperetin" is misspelled as "Hesperein" in multiple instances. Ensure consistent and correct spelling throughout the manuscript.
Response 3: Thanks for the advice. This has been corrected.
Comment 4: The reference formatting is inconsistent. Please revise all references to comply with the journal's guidelines.
Response 4: The reference section has been updated.
Comment 5: The manuscript lacks detailed nanoparticle synthesis parameters, including encapsulation efficiency (EE) and loading capacity (LC) values. These should be provided to enhance the credibility and reproducibility of the experiments.
Response 5: This information has been added in the result section, section 3, (Lines: 259-262). A paragraph is also added in the discussion section, section 4, Lines:424-427.
Round 2
Reviewer 1 Report
Comments and Suggestions for Authors
The authors replied well to comments.
Reviewer 2 Report
Comments and Suggestions for Authors
All the revisions provided were addressed accordingly. Now, the manuscript is suitable to be accepted for publication in Nanomaterials.